# Is Lung Ultrasound Helpful in COVID-19 Neonates?—A Systematic Review

**DOI:** 10.3390/diagnostics11122296

**Published:** 2021-12-08

**Authors:** Emil Robert Stoicescu, Ioana Mihaiela Ciuca, Roxana Iacob, Emil Radu Iacob, Monica Steluta Marc, Florica Birsasteanu, Diana Luminita Manolescu, Daniela Iacob

**Affiliations:** 1Department of Radiology and Medical Imaging, ‘Victor Babes’ University of Medicine and Pharmacy Timisoara, Eftimie Murgu Square No. 2, 300041 Timisoara, Romania; stoicescu.emil@umft.ro (E.R.S.); roxana.iacob@umft.ro (R.I.); fbirsasteanu@yahoo.com (F.B.); dmanolescu@umft.ro (D.L.M.); 2Research Center for Pharmaco-Toxicological Evaluations, ‘Victor Babes’ University of Medicine and Pharmacy Timisoara, Eftimie Murgu Square No. 2, 300041 Timisoara, Romania; iacob.daniela@umft.ro; 3Pediatric Department, ‘Victor Babes’ University of Medicine and Pharmacy Timisoara, Eftimie Murgu Square No. 2, 300041 Timisoara, Romania; 4Department of Pediatric Surgery, ‘Victor Babes’ University of Medicine and Pharmacy, Eftimie Murgu Square 2, 300041 Timisoara, Romania; radueiacob@umft.ro; 5Pulmonology Department, ‘Victor Babes’ University of Medicine and Pharmacy, Eftimie Murgu Square 2, 300041 Timisoara, Romania; marc.monica@umft.ro; 6Center for Research and Innovation in Precision Medicine of Respiratory Diseases (CRIPMRD), ‘Victor Babeș’ University of Medicine and Pharmacy, 300041 Timișoara, Romania; 7Department of Neonatology, ‘Victor Babes’ University of Medicine and Pharmacy Timisoara, Eftimie Murgu Square No. 2, 300041 Timisoara, Romania

**Keywords:** lung ultrasound, neonates, newborns, COVID-19, SARS-CoV-2

## Abstract

Background: The SARS-CoV-2 infection has occurred in neonates, but it is a fact that radiation exposure is not recommended given their age. The aim of this review is to assess the evidence on the utility of lung ultrasound (LUS) in neonates diagnosed with COVID-19. Methods: A systematic literature review was performed so as to find a number of published studies assessing the benefits of lung ultrasound for newborns diagnosed with COVID and, in the end, to make a comparison between LUS and the other two more conventional procedures of chest X-rays or CT exam. The key terms used in the search of several databases were: “lung ultrasound”, “sonography”, “newborn”, “neonate”, and “COVID-19′. Results: In total, 447 studies were eligible for this review, and after removing the duplicates, 123 studies referring to LU were further examined, but only 7 included cases of neonates. These studies were considered for the present research paper. Conclusions: As a non-invasive, easy-to-use, and reliable method for lung lesion detection in neonates with COVID-19, lung ultrasound can be used as a useful diagnosis tool for the evaluation of COVID-19-associated lung lesions. The benefits of this method in this pandemic period are likely to arouse interest in opening new research horizons, with immediate practical applicability.

## 1. Introduction

From the beginning of the pandemic until now, over 250 million cases of SARS-CoV-2 infections have been reported, according to the Worldometer website [1]. Out of these, a limited number of approximately 15% of cases are infections among children, infants, and newborns, according to the latest data provided by the American Academy of Pediatrics and the Children’s Hospital Association, in agreement with the data provided by the CDC. As their results indicate, approximately 7.8% of the newborns from tested COVID-positive mothers showed a positive PCR test at birth [2].

These small numbers of neonates with COVID-19 infection were confirmed by a study from Wuhan, China, where only three cases were reported out of a total of 33 pregnant women, with an infection rate of 9.09% [3]. In most of the cases, the newborns developed an asymptomatic or mild form of infection [4].

It is a fact that lung ultrasonography (LUS) has the great advantage of being easy to use; it is safe and non-invasive for infants, and at the same time, it is a useful method in the management of pulmonary and cardiac diseases in neonates. Lung artifacts detected by ultrasonography vary slightly with age, so it may be suitable for neonatal respiratory pathology, especially in the SARS-CoV-2 infection [5,6,7].

A number of studies demonstrate the usefulness of LUS in the evaluation of COVID -19-associated lung lesions in children, indicating that LUS sensitivity was higher than in the case of CT and CXR. Even if CT images present the highest accuracy, CT should not always be recommended for all children with COVID due to its significant irradiation potential. Furthermore, recent studies demonstrated a strong correlation between LUS and CT scores, proving that an ultrasound score higher than 20 (using 12-area score) is associated with an increased risk of complications and death [8,9]. As COVID-19 has become a worldwide disease, in many hospitals, the availability of CT scans for many patients diagnosed with COVID-19 is not widespread. Chest X-rays are more readily available, but they lack the necessary specificity and sensitivity for detecting COVID-19 pneumonia so as to be considered a reliable substitute method for CT scans [10]. Recent studies show the usefulness of LUS in detecting COVID-19, especially for pneumonia detection [11,12].

The findings concerning lung ultrasound procedures in the diagnostic assessment of children with COVID pneumonia are similar to the ones described in the case of adults and children with other types of pneumonia [13,14].

Lung ultrasound showed its utility in many other neonatal respiratory diseases, such as respiratory distress syndrome, transient tachypnea of the newborn, pneumonia, meconium aspiration syndrome, pulmonary hemorrhage and atelectasis of the newborn, pneumothorax, and so on [15,16]. Therefore, the aim of this review is to evaluate the utility of LUS for newborns diagnosed with COVID-19.

## 2. Materials and Methods

For the current literature review, the PubMed and ScienceDirect databases were searched, with a focus on articles written in English between December 1st 2019 and November 4th 2021. The articles were selected by searching the PRISMA tool (preferred reporting items for systematic reviews and meta-analyses), and in accordance with this, a systematic literature review was conducted. A total of 81 articles published between December 1st 2019 and November 4th 2021, covering the COVID period, were found involving the MeSH Terms ((“Ultrasonography”[Mesh]) AND “Lung”[Mesh]) AND “Infant, Newborn”[Mesh]. After this process, only seven articles were included and discussed in this study, according to the below-mentioned criteria.

The articles under scrutiny were selected according to the following criteria:-Pathology: COVID-19 pneumonia, SARS-CoV-2 infection;-Intervention/tool: lung ultrasound;-Age/population of interest: newborns, neonates, infants, <first 28 days of life.

Inclusion criteria:Original articles, special articles, systematic reviews, case reports, and scientific letters were considered;Studies involving a number of COVID-positive subjects were also included, as well as studies with only some of the participants being newborns;Only studies involving the lung/chest/thoracic ultrasound method were included;There were no exclusions regarding the language of the articles. One article written in Chinese with only the abstract in English was also included.

Only six published articles met the required criteria, and because it was found that the subject is scarcely addressed in the literature, it was necessary to perform two additional advanced searches on PubMed and ScienceDirect. A total of 133 articles from the reviewed period were found on PubMed, while 233 publications were found on ScienceDirect by searching with the following keywords: “lung”, “ultrasound”, “sonography”, “newborn”, “neonate”, and “COVID” (Figure 1—PRISMA flowchart). After this extended search, another study was found and was also included in the review. The flowchart below shows the steps that have been followed and the methodology used in deciding upon the eligible articles. From a total of 447 studies initially found, we removed duplicates, and we manually excluded articles with no ultrasound references. Ultimately, articles with no newborn mention were also excluded.

## 3. Results

After our search, we found and analyzed seven articles discussing the lung ultrasound method in newborns diagnosed with the SARS-CoV-2 infection, with a total of 58 subjects. The baseline characteristics of the patients were heterogenous; therefore, we underlined the most relevant ones, where data were available. The gender distribution was 41.93% males and 58.06% females, respectively. Regarding the gestational age, 15% of the neonates were preterm, while 85% were born on term. The infection was detected using the Reverse transcription polymerase chain reaction (RT-PCR) in 61.3% of the cases, while for the other subjects, the IgM tests were used. The age (days of life) of the patients when the COVID-19 infection was confirmed varied as follows: two studies with 3.8 ± 5.2 (mean ± standard deviation), another two studies with the median age of 8 days of life, and one study with the age between 1 and 18 days.

However, with the knowledge gained from the other articles based on this imagistic tool used in the evaluation of neonatal respiratory pathology, we can outline a number of further research directions (Table 1—characteristics of the articles included in the study).

### 3.1. LUS Artifacts

The main changes described after performing ultrasound procedures in the lung fields of newborns were:Decrease in and disappearance of A-lines;Sparse B-lines;Confluent B-lines;Abnormal pleural lines;Subpleural consolidation (Table 2—lung ultrasound findings and prevalence of the main changes from included studies).

#### 3.1.1. A-Lines Disappearance

The most common ultrasonographic changes found in the analyzed studies were the variations of physiological A-lines from decrease to disappearance, varying between 62.8% and 100% [17,18,22]. A-lines are described as hyperechogenic, horizontal, and equidistant ones that are parallel to the pleura [6]. These are the reflections of the ultrasound waves as they occur as an echo between the pleura and the transducer, similar to a reverberation artifact [6]. These variations of A-lines with some sparse or confluent B-lines can be considered the first sign of an alveolar-interstitial pattern [5].

#### 3.1.2. B-Lines

Sparse B-lines were the next most frequent modification with an occurrence rate between 55.3% and 100%, reported in four reviewed studies [17,18,19,22].

#### 3.1.3. Abnormalities of Pleural Lines

Abnormal pleural lines, such as thickening, interrupted, or irregular pleural lines, were reported differently in the referenced studies, ranging between 21.9% and 100% [17,18,19,22].

#### 3.1.4. Consolidations

Subpleural consolidation, recognized as a sign of viral pneumonia in association with variation of B-lines, were reported only in a few cases (between 1.5% and 66.6%) in the lung ultrasound exam in the target group [17,18,19,20,21,22]. No pleural effusion or pneumothorax was reported.

The predominant lesion distribution was bilateral in the inferior lobes and posterior segments, where the highest scores were recorded (right inferior posterior regions with a score 1.27 ± 0.47) [17].

Wei Li et al. found that most of the neonates included in their study had increased levels of creatine kinase myocardial band (CK-MB) but with no echocardiographic abnormalities [17].

### 3.2. LUS Scores in Neonates with COVID-19 Pneumonia

Only three articles used a score for the semi-quantitative analyses of the lung lesions. Two studies indicated a 12-area score, while the other one adopted a score with just six segments (anterior, lateral and posterior for each hemithorax) [17,18,19]. The 12-area score was determined by Mongodi et al., with six regions per hemithorax, the axillary lines as landmarks, placed in anterior and posterior positions, and using the nipple line as a demarcation mark between the upper and lower regions [23].

Two out of the three studies were located in Wuhan, the epicentre of the COVID-19 pneumonia. One compared 11 neonates with the SARS-CoV-2 infection (two preterm and nine babies at term, with asymptomatic or mild forms) with a healthy control group similar in characteristics (age, gender). Lung ultrasound score was higher in the infection group (8.4 ± 1.7 vs. 2.3 ± 1.4), and the most frequent lesions were erased A-lines and sparse/confluent B-lines. Additionally, it is important to note that abnormalities were identified by ultrasound in three of the infected newborns with previous normal radiographic exams. The authors concluded that all pulmonary injuries were bilateral with multiple areas affected, specifically located in the lower lobes and in the middle lobe [17,18].

The other study, which was conducted in Wuhan, China, examined a number of five cases presenting gastrointestinal and respiratory symptoms. In all these cases, the pulmonary ultrasonography showed pleural line abnormalities and pulmonary edema in different conditions, described as confluent B-lines with an interstitial pattern. Additionally, some small areas with pulmonary consolidation were visible. A very important fact to remember is that lung ultrasound has a higher sensitivity rate compared with chest X-ray and thoracic computer tomography in the diagnosis of pulmonary edema [22].

Another article presents the evolution of three neonates during their admission to a neonatology department in Toledo, Spain. They were examined every 48 h using the ultrasound technique, and the cases were graded by severity following a three-area score for each hemithorax. The highest recorded score was 10/18 in a newborn with severe hypoxic-ischemic encephalopathy and meconium aspiration. These newborns presented a number of comorbidities, such as meconium aspiration pneumonia, bronchopulmonary dysplasia, and Hirschsprung’s disease, shown as the limitations of the study. They compared the findings with a previous LU performed in a neonate with bronchopulmonary dysplasia, but with no following results. All neonates described in this report had a mild form of respiratory infection [19].

A significantly larger studied group included 27 neonates from Sao Paulo, Brazil. The study presents the imagistic aspects in patients who tested positive for SARS-CoV-2 or who were highly suspected to be infected. The subjects were divided into two groups, one with respiratory symptoms and the other with no pulmonary manifestations. In the COVID-19 group with respiratory symptoms were reported several coalescent B-lines, pleural irregularities, and some subpleural consolidation [20].

Pineda Caplliure et al. examined a case confirmed with COVID-19 pneumonia that presented a fluctuant evolution with a depreciation at 24 h after admission. The newborn was examined by echocardiography, chest X-ray, and lung ultrasound, which showed the specific subpleural and pleural changes with variation of B-lines [21].

Most of the newborns were asymptomatic or presented mild forms of infection, only 20% of the subjects presenting fever. Additionally, respiratory, gastrointestinal, and cardiac manifestations accompanied the symptomatic forms of the infection. The comorbidities were reported in one study with the above-mentioned examples (Hernández et al.) [19].

## 4. Discussion

Following the evaluation of the above-mentioned articles, it can be noted that a small number of studies discusses in depth the ultrasound method as a useful instrument of exploration in COVID-positive newborns, a fact that is also confirmed by Caroselli et al. in their review, which found only seven articles about lung ultrasound at newborns, infants, and children with SARS-CoV-2 infection [7].

There is a growing trend in the use of pulmonary ultrasonography, especially in newborns, as evidenced by the numerous recent articles using this imagistic technique, if we compare with the results obtained by conventional radiography or the CT exam [7,24].

Jones et al. note a significant decrease of about 38.8% in the use of chest X-ray procedure, but with very good results in detecting pneumonia with the aid of lung ultrasound. This indicates that ultrasonography is a suitable diagnosis tool that can replace the other more conventional methods that are likely to be more harmful and potentially more invasive [7,25]. Additionally, Feng et al. prove that chest ultrasound has a higher sensitivity and accuracy rate compared to X-ray and CT procedures in the diagnosis of pulmonary edema, suggesting that ultrasonography could be used in the monitoring and evaluation of the lung injuries caused by SARS-CoV-2 infection [22]. LUS findings are directly correlated with CT score at intensive care unit admission and inversely correlated with blood test results (PaO_2_, C-reactive protein) [9,26]. Other studies following a similar research line demonstrate that LUS has a higher specificity and sensitivity rate in detecting consolidations, interstitial and alveolar syndromes, and pleural effusion, specific for a pediatric pneumonia, compared to conventional chest radiography [27,28,29].

Berce et al. underline the importance of LUS as a useful method in a bacterial or viral outbreak or in epidemic times [30]. Lately, several studies have shown a significant role of LUS in the evaluation of COVID-19 pneumonia, not only regarding its accuracy in the detection of lesions, but also in evaluating the risk of death and complications development [8,31].

The majority of patients examined in the referenced studies were asymptomatic or had a mild form of infection, causing pulmonary and gastrointestinal symptoms, but also an unusual symptomatology was described, such as erythema [32]. According to Lingkong Zeng et al., the most frequent manifestation was dyspnea, present in a number of four neonates, out of a total of 33 cases [3]. The pulmonary symptoms were mentioned by Li et al. in their 11-cohort group, accompanied by digestive complaints [17]. Additionally, Feng et al. report the same manifestations [22]. Both the respiratory and digestive manifestations can be determined by ultrasonography [6,33,34].

Another system with modified parameters was the cardiac one, with reported elevated CK-MB levels and the suspicion of myocardial injury or dysfunction [3,18,35]. The cardiac function was examined using echocardiography, which showed normal parameters in all cases [21,35]. Taking into consideration that all patients are clinically vulnerable and the long-term effects of the SARS-CoV-2 infection are yet to be discover, it would be necessary to have an additional investigating tool, such as the genetic expression of miRNAs, with a higher selectivity and specificity rate than the usual biomarkers [36]. For a better analysis of myocardial dysfunctions, echocardiogram and echocardiography have proven effective, as they are used in the ‘SAFE protocol’ for critically ill neonates [5,37,38].

According to Volpicelli and Francisco, an interstitial pattern with variation of B-lines with occasional spared areas are indicative of a viral pneumonia. The studies included in this article describe similar changes in lung parenchyma with sparse B-lines as a typically detectable lesion [39]. These changes emerging from viral infections are identified in the findings following CT examinations of the above-mentioned virus infection [7,40,41]. One study concluded that LUS detected all the COVID-19 pneumonia cases compared to the CT exam, with a sensitivity and specificity of 100% and 78.6%, respectively [42].

The peripheral, subpleural area was the front-runner for the ultrasound changes visualized during the SARS-CoV-2 infection due to the small size of the viral particles that easily reach the distal bronchioles and alveoli [17]. The pathological modifications were seen as consolidation spread around the pleura due to the impacting of the alveoli with mucoid material, especially in an immature lung [43,44]. The results of the studies analyzed describe the same topography with a predominance of the lesion in the lower lobes and in the posterior segments [45]. Likewise, Mento et al. conclude that the posterior zones must be explored in all patients with COVID-19 pneumonia, being the most affected area, and they also recommend a 12-area ultrasound examination [46]. In addition, Smargiassi et al. propose a multiple area approach for the lung ultrasound exam [47].

Decrease or disappearance of A-lines is the most common pattern detected by lung ultrasonography, according to the above-mentioned studies, and it is also the first pulmonary change that appears in alveolar–interstitial pathologies [5,48]. For example, Li et al. found variation in A-lines that were even disappearing in the 83 analyzed regions out of a total of 132 regions, based on their 12-area score. It is Feng et al. who discovered these variations in all neonates, but without previously dividing the lung in multiple zones [17,18,22].

On the other hand, ‘here to there’ B-lines, described as sparse B-lines, were present in varying degrees in all patients examined by lung ultrasonography, ranging between 55.3% and 66.6% [17,18,19]. These facts are significantly consistent with the results presented by Caroselli et al. in their review of the neonates, children, and adolescents enrolled in their study, indicating the presence of B-lines in a proportion of 50% [7]. These changes result from a fluid collection that expands the interlobular septs and is usually found in pulmonary edema and pneumonia [6,49,50]. Coalescent or confluent B-lines and ‘white lung’ symptoms were identified in a quarter of the patients [7], but in the studies included in our review, the figures differ from study to study (7.6%, 66%, and 100%, respectively). The higher values were presented by the studies with smaller batches of patients (three and five newborns). For this reason, these results cannot be reproduced and integrated with other pediatric and literature data available [10,51,52,53,54].

Another controversial subject concerns the differences between subpleural consolidations and their occurrence rate. Hernandez et al. had the highest percentage of this type of lesions (12/18 affected regions—66.6%), because they examined only three patients in evolution, unlike the other authors that included more subjects in their studies. Therefore, the probability of finding subpleural consolidation is quite low [19]. Thus, the follow-up of the infected neonates can reveal advanced modifications, such as lung consolidations [19].

At the beginning of the COVID-19 pandemic, irregularities and thickenings of the pleura were among the first signs described by lung ultrasonography, accompanied by subpleural consolidations [12]. Abnormalities of the pleura are the hallmarks described by all the reviewed authors discussing pulmonary ultrasound, with a range between 21.9% and 44.4%. The values generated by Caroselli et al. are in the middle of those reported (34.09%) [7,17,18,19]. This alteration of the pleura is also seen in community-acquired pneumonia [55,56].

Pleural effusions have not been found in SARS-CoV-2 infections, but only in exceptional cases. No pleural effusions were identified in any of the original articles included in this review, while Caroselli et al. report a rate of 6.82% showing the use of LUS. This can be due to the fact that ultrasound is able to detect a small effusion [6,57]. Additionally, the pleural effusion in COVID-19 pneumonia is a rare finding [58].

Given the fact that CT and chest X-ray exams produce irradiation and are harmful, with long-term side effects, especially for newborns, a radiation-free, non-ionizing, real-time, user -riendly, and repeatable tool is necessary [59,60,61]. Moreover, during this pandemic, hospital circuits have been completely changed, making it more difficult for patients to be transported to radiology departments to be examined by conventional radiography or CT [51]. Furthermore, according to the statistics presented by the World Human Organization, around two-thirds of the world population does not have easy access to a basic radiology or imagistic department [7]. Additionally, ultrasound equipment is readily available in many hospital departments, wards, and small clinics [62]. Unfortunately, this is not the case with radiographs or CTs.

Recently, LUS has proved its efficiency in the dynamic monitoring of COVID-19 pneumonia, as well as in the follow-up of patients recovered from this infection [63,64]. Moreover, a recent review proves the usefulness of hand-held ultrasound equipment, which has the advantage of lower acquisition costs and that is easy to decontaminate, which is a necessary activity nowadays [65]. Lung ultrasonography is the answer to all the demands that have been formulated in the course of this article. Once the lung ultrasound is performed, the procedure could be also used for examining other systems affected by COVID-19, such as the cardiac and gastrointestinal ones.

We must acknowledge the limitation of this review, as only seven studies were reported in our area of interest, with only a small sample size of patients and a lack of data homogeneity. Further multicentric studies are likely to enhance the accuracy of the research results and other important LUS application, as several studies have already showed significant progress in adults and children with COVID-19 pneumonia.

## 5. Conclusions

LUS proves to be useful in the detection of lung injury in the studies that were reviewed, even if only a small number of publications related to pulmonary ultrasonography in neonates with COVID pneumonia were found. Considering the advantages of lung ultrasound, as well as its utility in this ongoing pandemic period, this method is likely to stir interest in opening new research horizons with increased practical applicability.

## Figures and Tables

**Figure 1 diagnostics-11-02296-f001:**
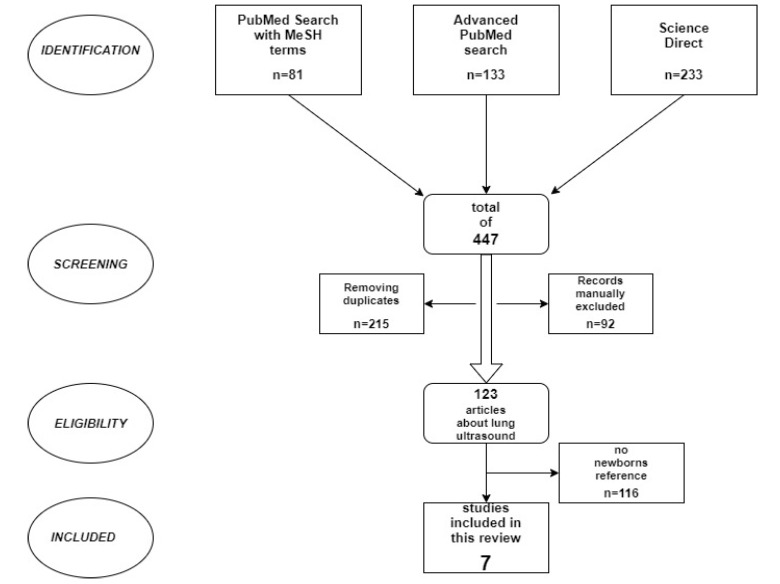
PRISMA flowchart. Literature review.

**Table 1 diagnostics-11-02296-t001:** Characteristics of the articles included in the study.

No.	Title	Authors	City and Country	Year of Publication	Type	No. of Cases	Observations
1	Quantitative assessment of COVID-19 pneumonia in neonates using lung ultrasound score [17]	Wei Li et al.	Wuhan China	2021	original article	11	
2	Bedside lung ultrasound score (LUSS) on assessing pneumonia in COVID-19 neonates [18]	Wei Li et al.	Wuhan China	2020	original article	11	
3	Point-of-care lung ultrasound in three neonates with COVID-19 [19]	R. Gregorio- Hernández et al.	Toledo Spain	2020	original article	3	
4	Use of lung ultrasound in neonates during the COVID-19 pandemic [20]	Marcia Wang Matsuoka et al.	Sao Paulo Brazil	2020	special article/original article	27	positive and negative COVID neonates
5	Usefulness of chest ultrasound in a neonatal infection due to SARS-CoV-2 [21]	Pineda Caplliure A et al.	Valencia Spain	2021	scientific letter	1	
6	Application of pulmonary ultrasound in the diagnosis of COVID-19 pneumonia in neonates [22]	X Y Feng et al.	Wuhan China	2020	original article	5	abstract in English, article in Chinese
7	Diagnostic Imaging in Newborns, Children and Adolescents Infected with Severe Acute Respiratory Syndrome Coronavirus 2 (SARS-CoV-2): Is There a Realistic Alternative to Lung High-Resolution Computed Tomography (HRCT) and Chest X-Rays? A Systematic Review of the Literature [7]	Costantino Caroselli et al.		2021	review	44	newborns, children and adolescents

**Table 2 diagnostics-11-02296-t002:** Lung ultrasound findings and prevalence of the main changes from included studies.

No.	Title	No. of Cases	Decreasing/Disappearing A-Lines (Total)	Sparse B-Lines	Confluent B-Lines	Abnormal Pleural Lines	Subpleural Consolidation	Predominant Lesions	Myocardial Injury	Observation
1	Quantitative assessment of COVID-19 pneumonia in neonates using lung ultrasound score [17]	11	83/132 regions 62.8%	73/132 regions 55.3%	10/132 regions 7.6%	29/132 regions 21.9%	2/132 regions 1.5%	bilateral inferior and posterior	The majority of patients	12 regions for neonate’s chest areas
2	Bedside lung ultrasound score (LUSS) on assessing pneumonia in COVID-19 neonates [18]	11	52 decreased 31 disappear 83/132 regions 62.8%	73/132 regions 55.3%	10/132 regions 7.6%	29/132 regions 21.9%	5/132 regions 3.8%	bilateral lower lobes and right middle lobe	-	12 regions for neonate’s chest areas
3	Point-of-care lung ultrasound in three neonates with COVID-19 [19]	3	12/18 regions 66.6%	12/18 regions 66.6%	12/18 regions 66.6%	8/18 regions 44.4%	12/18 regions 66.6%	bilateral posterior areas	-	6 regions for neonate’s chest areas
4	Use of lung ultrasound in neonates during the COVID-19 pandemic [20]	27	Present	Present	Present	Present	Present	posterior lung fields	-	positive and negative COVID newborns
5	Usefulness of chest ultrasound in a neonatal infection due to SARS-CoV-2 [21]	1	Present	Present	Present	Present	Present	bilateral posterior areas	-	-
6	Application of pulmonary ultrasound in the diagnosis of COVID-19 pneumonia in neonates [22]	5	5/5 100%	5/5 100%	5/5 100%	5/5 100%	1/5 20%	-	-	-
7	Diagnostic Imaging in Newborns, Children and Adolescents Infected with Severe Acute Respiratory Syndrome Coronavirus 2 (SARS-CoV-2): Is There a Realistic Alternative to Lung High-Resolution Computed Tomography (HRCT) and Chest X-Rays? A Systematic Review of the Literature [7]	44	Present	50%	25%	34.09%	43.18%	posterior lower lung fields	-	review with newborns, children and adolescents

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
