# Peer review of "Is Lung Ultrasound Helpful in COVID-19 Neonates?—A Systematic Review"

_diagnostics, 2021, doi:10.3390/diagnostics11122296_

Round 1
Reviewer 1 Report
I read with interest the paper of these authors that aimed at evaluating the role of lung uitrasound in neonates with COVID-19. This paper could have potential of interest for readers, the use of lung ultrasound in neonates with COVID-19 is still unclear. However, I think that authors should resolve several major issues before considering this paper for publication.
1) English language should be extensively revised (and certified) by a native-expert speaker. Several sentences are not clear and significance could be misinterpreted.
2) Discussion is superficial, the authors do not report the recent evidence about diagnostic accuracy of LUS in COVID-19 patients. In particular, the prognostic role of LUS should be discussed, recent data show that a high LUS score (greater than 20) could be associated with worse prognosis and risk of death [Hernández-Píriz A, et al. Usefulness of lung ultrasound in the early identification of severe COVID-19: results from a prospective study. Med Ultrason. 2021 Nov 10. doi: 10.11152/mu-3263.
Tana C, et al. Prognostic Significance of Chest Imaging by LUS and CT in COVID-19 Inpatients: The ECOVID Multicenter Study. Respiration. 2021 Aug 31:1-10. doi: 10.1159/000518516. Epub ahead of print. PMID: 34515247; PMCID: PMC8450833.
Orlandi D, et al. Coronavirus Disease 2019 Phenotypes, Lung Ultrasound, Chest Computed Tomography and Clinical Features in Critically Ill Mechanically Ventilated Patients. Ultrasound Med Biol. 2021 Dec;47(12):3323-3332. ].
3) A meta-analysis of the included studies is high recommended.
4) The authors should discuss more about limitation of this systematic review (small sample size of patients evaluated in the single studies, heterogeneity of data, etc)
5) I suggest to report more data about the baseline characteristics, comorbidities (e.g. other lung disorders and autoimmunity) of patients included in the studies
Reviewer 2 Report
This paper provide a systematic review of Ultrasound in COVID-19 Neonates.The objective function studied in this paper is to The aim of the review is to assess the evidence on the utility of lung ultrasound (LUS) in neonates diagnosed with COVID-19. A new metaheuristic algorithm is proposed to address this problem. There are a few weaknesses that should be addressed in this paper. Therefore, I suggest the authors resubmit it after a minor revision. My suggestions are as follows: As the first step, I strongly suggest that the paper be proofread and reread meticulously again, particularly in regard to the spelling and grammatical mistakes. The paper should be revised to include more recent references on the suggested topic. At least you need 60 references for a review paper. However, you consider only 52 references. It is necessary to include additional information in section 1 or the introduction in order to define the suggested review. Please elaborate on the flowchart you visually presented in figure 1, and provide additional explanation. You should provide documentation that supports evidence for why you believe these 7 suggested papers after filtering are valuable. Although the flowchart is beneficial, It’s also important to outline the methodology behind your recommendation.
Round 2
Reviewer 1 Report
the authors have resolved the issues